# Emergence of Abstract State Representations in Embodied Sequence Modeling

**Tian Yun**[♠][*]   **Zilai Zeng**[♠][*]   **Kunal Handa**[♠]   **Ashish V. Thapliyal**[♠]   **Bo Pang**[♠]

**Ellie Pavlick**[♠]   **Chen Sun**[♠]

[♠]Brown University, [♠]Google Research

{tian_yun, zilai_zeng, chensun}@brown.edu

## Abstract

Decision making via sequence modeling aims to mimic the success of language models, where actions taken by an embodied agent are modeled as tokens to predict. Despite their promising performance, it remains unclear if embodied sequence modeling leads to the emergence of internal representations that represent the environmental state information. A model that lacks abstract state representations would be liable to make decisions based on surface statistics which fail to generalize. We take the BabyAI environment, a grid world in which language-conditioned navigation tasks are performed, and build a sequence modeling Transformer, which takes a language instruction, a sequence of actions, and environmental observations as its inputs. In order to investigate the emergence of abstract state representations, we design a "blindfolded" navigation task, where only the initial environmental layout, the language instruction, and the action sequence to complete the task are available for training. Our probing results show that intermediate environmental layouts can be reasonably reconstructed from the internal activations of a trained model, and that language instructions play a role in the reconstruction accuracy. Our results suggest that many key features of state representations can emerge via embodied sequence modeling, supporting an optimistic outlook for applications of sequence modeling objectives to more complex embodied decision-making domains.[1]

## 1 Introduction

Sequence modeling, the task of predicting the next symbol based on the given context (e.g., words, pixels), is a simple yet versatile objective. Leveraging this sequence modeling objective and the Transformer (Vaswani et al., 2017), language models have achieved impressive capabilities on various tasks, such as code writing (Chen et al., 2021b), mathematical problem solving (Lewkowycz et al., 2022), and question answering (Sanh et al., 2022).

Motivated by the success of sequence modeling in language models, decision making via sequence modeling treats each action of an embodied agent as a token, and formulates problems traditionally solved via reinforcement learning (RL) as a sequence modeling problem (Chen et al., 2021a; Janner et al., 2021). Although these models show impressive performance, it remains unclear whether such embodied sequence modeling leads to the emergence of internal representations that encode the structure of the external environment, or rather that the models simply learn to predict next actions based on "surface statistics" – correlations that encode the likelihood of an action based on the given contexts. If the latter is true, we would need to re-examine the underlying reasons for the strong empirical success of the sequence models. However, if the sequence models do encode an implicit world model, it can help us better understand their effectiveness for decision making tasks.

In this work, we study whether abstract state representations emerge in embodied sequence modeling. We use the BabyAI (Chevalier-Boisvert et al., 2019) environment to design a "blindfolded" navigation task, where only the initial layout of the environment, a language instruction, and an action sequence which completes the task are available during model pretraining. This agent can follow the instructions by memorizing the action sequences it has seen during pretraining, or by constructing and updating abstract environmental representations and then picking next moves based on these representations. Although the model has no prior knowledge about the intermediate layouts of the environment (i.e., non-initial layouts), our probing results demonstrate that a trained model learns internal representations of the intermediate layouts.

---

[1]Project webpage and other resources are available at: https://abstract-state-seqmodel.github.io

*: First two authors contribute equally. Part of the work was performed while TY was a student researcher at Google.

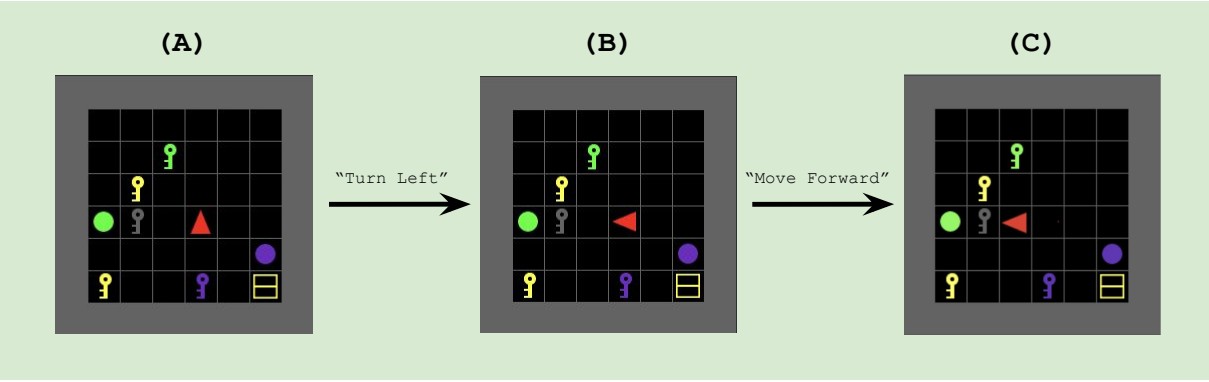

Figure 1: A demonstration of a language-conditioned navigation task in BabyAI *GoToLocal* level (Chevalier-Boisvert et al., 2019) where an agent (i.e., red triangle) needs to navigate to the target objects specified in the given instructions. This is an example trajectory where the agent achieves the goal of "go to the grey key": (A) The board is randomly initialized with the agent and 8 objects. (B) The agent navigates in the environment to try to approach the target object. (C) The agent keeps navigating until it reaches the target object.

In this navigation task, there are three components – layouts of the environment, language instructions, and action sequences – each of which can impact the emergence of abstract state representations in embodied sequence modeling. We further raise the question: what role do language instructions play in the emergence of such representations? We ablate with regards to the presence or absence of language instructions during model training, and train probes to recover the intermediate environmental layouts. We observe that language instructions indeed facilitate the emergence of abstract environmental representations.

To sum up, we present three contributions: (1) we demonstrate that abstract state representations do emerge via embodied sequence modeling in the BabyAI navigation task; (2) we study the impacts of language instructions in the emergence of such representations, and find that language instructions can strengthen such emergence in embodied sequence modeling; (3) we show that a model trained with only the initial environment layout learns internal representations of the environment and can perform competitively against a model trained with all the available information, which supports the promise of sequence modeling objectives for decision making problems.

## 2  Related Work

**Decision Making via Sequence Modeling.** Inspired by the success of sequence modeling in language models, some recent work (Janner et al., 2021; Chen et al., 2021a; Li et al., 2022; Zeng et al., 2023; Wu et al., 2023) formulates decision making problems using sequence modeling by treating actions as tokens. Although decision making via sequence modeling has shown impressive performance on downstream tasks, sequence modeling objective does not explicitly drive the models to learn abstract and updatable representations of environments, which have been shown to be essential for solving decision making problems (Yang and Nachum, 2021; Hafner et al., 2020). Thus, in order to further understand why decision making via sequence modeling is successful despite minimal inductive biases, in this work we examine whether the sequence modeling objective leads to the emergence of such representations of environments.

**Probing Transformer-based architectures.** Though sequence modeling has been shown to be successful in language modeling (Raffel et al., 2020; Sanh et al., 2022), image modeling (Dosovitskiy et al., 2021), and decision making (Janner et al., 2021; Li et al., 2022; Pashevich et al., 2021), it remains unknown whether such training results in models that encode rich abstractions or rather simply model surface statistics of their domains. Probing is one standard tool (Tenney et al., 2019; Li et al., 2023; Alain and Bengio, 2016) used to understand the internal representations of neural networks. A probe is a classifier that is trained to predict some salient features from the internal activations of a model. If a trained probe can accurately predict a feature of interest, it suggests

that a representation of this feature is encoded in the internal activations of this model.

Such work has suggested that sequence modeling of language tokens leads to representations of abstract concepts such as syntax (Hewitt and Liang, 2019; Tenney et al., 2019) and even non-linguistic notions such as of colors and directions (Patel and Pavlick, 2021; Abdou et al., 2021). Such findings suggest that sequence modeling over embodied domains (like we study here) could similarly result in abstractions over states and actions. We utilize probes to recover the intermediate layouts of the environment to understand whether decision making via sequence modeling can automatically learn grounded representations of the environments.

Othello-GPT (Li et al., 2023) shows that a GPT (Radford et al., 2019) learned to model Othello game moves learns internal representations of board states. That work assumes fixed initial layout, which is different from most real-world tasks. In our work, we use the BabyAI (Chevalier-Boisvert et al., 2019) environment, which considers random initial layouts, to investigate whether sequence modeling leads to emergence of grounded representations of environments.

## 3 Embodied Sequence Modeling

### 3.1 Problem Setup

In this work, we consider the following scenario: given a task specified by natural language instructions, an agent needs to predict next actions based on past experiences (e.g. previous states, actions taken) to complete the task. This process can be formulated as embodied sequence modeling, where each input (e.g. states, actions, language instructions) is represented in the form of tokens. The sequence first defines a task via a language instruction and the initial environmental layout. During training, it then demonstrates how the task can be completed via a sequence of actions (and optionally the corresponding intermediate states). Formally, we define embodied sequence modeling as next (action) token prediction task:

$$\max_{\theta} \log P_{\theta}(a_t | g, s_t, h_{t-1})$$

which aims to maximize the likelihood of the next action token $a_t$ given the state token at current timestep $s_t$, language instruction tokens $g$ and past experiences $h_{t-1} = \{s_i, a_i\}_{i=1}^{t-1}$. We train our sequence modeling framework by optimizing the

above objective over a dataset $\mathcal{D} = \{(\tau^k, g^k)\}_{k=1}^N$, which holds a set of (trajectory, goal) pairs. Each trajectory $\tau^k$ includes a sequence of states $\{s_i\}_{i=1}^T$ and actions $\{a_i\}_{i=1}^T$ that records how the agent successfully reaches the corresponding goal $g^k$, where $T$ is the length of a trajectory.

### 3.2 BabyAI

We seek to understand whether embodied sequence modeling leads to the emergence of abstract internal representations of the environment. We use the BabyAI (Chevalier-Boisvert et al., 2019) environment as a testbed to investigate this. The goal in this environment is to control an agent to navigate and interact in an $N \times N$ grid to follow the given language instructions (e.g., "go to the yellow key"). The state of BabyAI is a symbolic observation of the grid, where each grid cell is represented by two attributes: the type and color of the object. At each timestep, the agent can take an action (e.g., "turn left", "move forward") based on the current state of the grid (Figure 1). Since this environment is synthetic, we can generate arbitrary amount of data and ensure that there is no overlap. This environment also has a fixed mapping between the intermediate layout and a pair of initial layout and action sequence, which facilitates our following use of probes to reconstruct the intermediate states.

### 3.3 Sequence Modeling Setups in BabyAI

In this section, we introduce two sequence modeling setups in BabyAI, namely the Regular setup and the Blindfolded setup.

**Regular setup:** The objective is to predict next actions based on all the information at the current and the previous steps provided explicitly including intermediate states $s_i$ which are computed using the BabyAI environment: $P(a_t | g, s_t, h_{t-1})$, where $h_{t-1} = \{s_i, a_i\}_{i=1}^{t-1}$. Note that $s_t$ is computed external to the sequence model by interacting with the BabyAI environment given $h_{t-1}$.

**"Blindfolded" setup:** The objective is to predict next actions based on the language instruction, prior actions, and only the initial state: $P(a_t | g, s_1, \{a_i\}_{i=1}^{t-1})$. In this case, the current state is not fed to the model explicitly. By comparing task completion performance of this setup against the regular setup, we hope to gain insights into whether these sequence models are able to internally reconstruct the intermediate states implicitly in order to determine the right next actions to take.

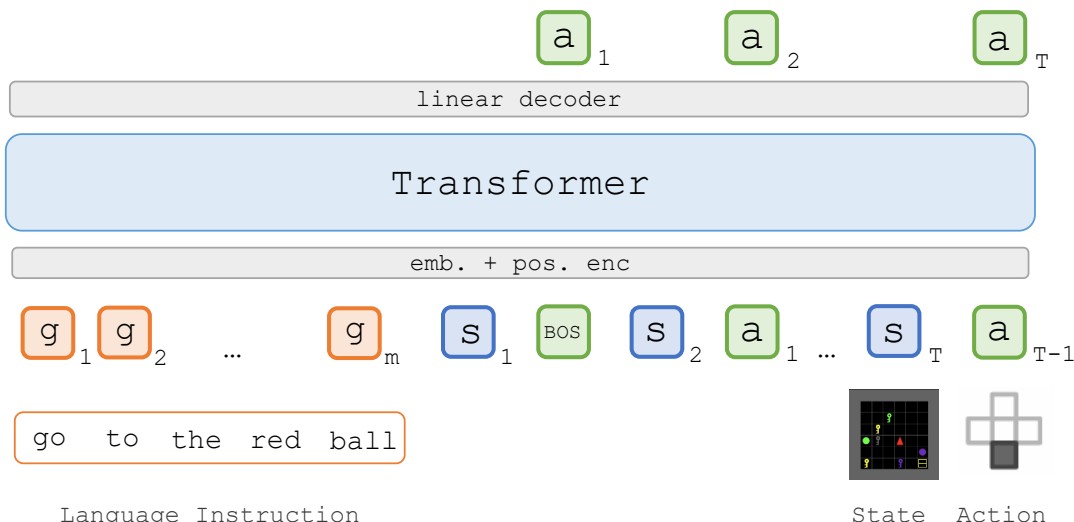

Figure 2: An overview of our model architecture. Our Transformer-based sequence model trains on a sequence of states, actions and language instruction tokens, with the objective of predicting next action tokens given past experiences and the language instruction.

## 4 Methods

### 4.1 Architecture and Model training

We build our embodied sequence modeling framework with a Transformer-based architecture, in which we can process heterogeneous data (e.g. natural language instructions, states and actions) as sequences and autoregressively model the state-action trajectories with causal attention masks.

Figure 2 illustrates the regular setup where the model is given the current state and the full history of state-action pairs, along with a natural language instruction, to predict the next action. We denote the input $\mathcal{I}$ to our model as a tuple $(g_{1:m}, s_{1:T}, a_{1:T-1})$ of the goal, state and action sequence, where the goal $g$ is composed of $m$ natural language tokens. Additionally, we prepend a [BOS] token to the action sequence to facilitate autoregressive action prediction:

$$\mathcal{I} = (g_1, .., g_m, s_1, [\text{BOS}], s_2, a_1, ..., s_T, a_{T-1})$$

To employ a neural network for sequence modeling, we need to first embed the language, state, and action inputs into the model's latent space of dimension $d$. We learn two embedding lookup tables to project the language tokens and action inputs into fixed-length vectors of dimension $d$, respectively. Each state observation $s$ is represented by $N \times N$ grid cells. Each cell is described by the discrete object type and color attributes, which are embedded separately and concatenated into a fixed-length

embedding vector. To encode a state, we apply a two-layer convolutional neural network with *same* padding, flatten the output feature map across all grid cells, and project it to a $d$-dimensional embedding with a 3-layer MLP. We use sinusoidal embedding to encode positional and temporal information for the language, action, and state tokens. Our sequence model, a Transformer with causal attention masks, is trained end-to-end by minimizing the cross-entropy loss between the predicted and ground-truth actions.

Prior works (Chen et al., 2021a; Janner et al., 2021) formulate sequential decision making as a sequence modeling problem. They train a policy with GPT architecture (Radford et al., 2019), which shares the same objective (i.e. next action token prediction) as ours. Thus, both regular and "blind-folded" sequence modeling (see Section 3.3) can serve as policies conditioned on natural language instructions, by setting $\mathcal{I}$ to include all the observed states and actions up to the current time step $t$. We can then evaluate the sequence models using the standard BabyAI decision making benchmarks.

### 4.2 Probes for Internal Representations of Environmental Layouts

Once we have models trained to perform a language-conditioned navigation task, we train simple classifiers as probes (Tenney et al., 2019; Alain and Bengio, 2016) to explore whether the internal activations of a trained model contain representa-

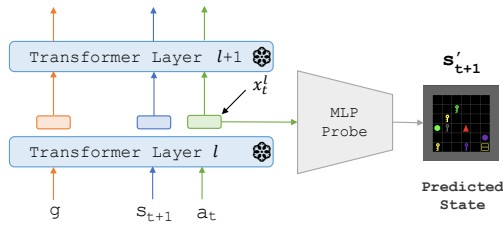

Figure 3: An overview of our probing pipeline. We train a 2-layer MLP probe to reconstruct intermediate states from the internal activations in our model.

tions of abstract environmental states (e.g., current board layouts). If the trained probes can accurately reconstruct the current state when it is not explicitly given to the model (e.g., in the "blindfolded" setup), it suggests that abstract state representations can emerge during model's pretraining. As illustrated in Figure 3, we denote the internal activations of the $t$-th action token after the $l$-th Transformer layer as $x_t^l$. During training, the internal activations are passed into a 2-layer MLP to reconstruct the state $s_{t+1}$, and the objective is to minimize the cross-entropy loss between the reconstructed and ground-truth states.

## 5 Experimental Setup

### 5.1 Pretraining Data & Model Variants

**Data generation.** We select *GoToLocal* (Figure 1) and *MiniBossLevel* (Figure A1 in Appendix) levels in BabyAI (Chevalier-Boisvert et al., 2019). The GoToLocal environment has a layout of $8 \times 8$ grids (including the walls) in a single room. Each task involves a single goal, namely navigating to the target object. To validate the generality of our observations, we also use the MiniBossLevel environment (see Figure A1 in the appendix), which has a layout of $9 \times 9$ grids separated into four rooms. The rooms are connected by lockable doors. The tasks are compositional, which may involve solving multiple sub-goals sequentially (e.g., toggle the door then pick up an object). In order to solve the tasks in both environments, the sequence model needs to ground the object descriptions in the language instructions into object attributes and their locations in the "grid-world" environment.

For both environments, we create 1M training trajectories respectively, each of which consists of a language instruction, a state sequence, and an action sequence. We use the BOT agent (Chevalier-

Boisvert et al., 2019) to generate trajectories. We select the trajectories that successfully solve the final tasks, and discard the rest.

**Complete-State Model** simulates the regular setup of sequence modeling in BabyAI (Section 3.3). Thus, this model predicts next actions based on all the information at current and previous steps: $P(a_t|g, s_t, h_{t-1})$, where $h_{t-1} = \{s_i, a_i\}_{i=1}^{t-1}$.

**Missing-State Model** simulates the "blindfolded" setup. Therefore, this model predicts next actions based on the language instruction, prior actions, and only the initial state: $P(a_t|g, s_1, \{a_i\}_{i=1}^{t-1})$. In practice, we observe this vanilla setup is too challenging as the model lacks sufficient information to understand the language instructions. We therefore choose to additionally provide the last state right before the agent solves the task for 50% of the time during training. During evaluation, we always only pass in initial states.

### 5.2 Probing Setup

**Data for probing.** To train the probes, we reuse the 1M trajectories used for pretraining the sequence models. We then collect 100K trajectories each for the validation and test sets to evaluate the trained probes. Following, we pass the trajectories of each split, paired with corresponding goals, to the pretrained Complete-/Missing-State models, and extract internal activations for action tokens. For each input, we randomly sample a timestep $t$ and retrieve $t$-th action token's internal activations $x_t^l$ in each Transformer layer $l$. To focus on whether Missing-State model learns internal representations of *intermediate* states, we only sample from $t > 1$ — otherwise the Missing-State model can directly access ground-truth state information through $s_1$.

**Probe metrics.** We design three types of metrics to cover different aspects of states: (1) *Agent-Loc*: Agent locations are represented as discrete *(x, y)* coordinates on the grid. Location prediction can thus be framed as classification (e.g. $6 \times 6$ classes for GoToLocal, since we ignore the walls), from which we report the classification accuracy. (2) *Board-Type/Color*: We measure the probing classifiers' performance on predicting the object type and color attributes for all cells on the grid. An empty cell has the object type *background*. GoToLocal has 6 types and 6 colors, while MiniBossLevel has 7 types (door is the extra object) and 6 colors. We report the mean average precision (mAP) over all

| | Random Init. | Agent-Loc | Board-Type | Board-Color | Neighbor-Type | Neighbor-Color |
|---|---|---|---|---|---|---|
| Initial State | n/a | 7.4 | 82.2 | 80.1 | 39.8 | 23.2 |
| Complete-State Model | ✓ | 10.2 | 27.4 | 21.2 | 47.7 | 31.7 |
| | | 97.8 | 56.6 | 48.5 | 84.4 | 83.4 |
| Missing-State Model | ✓ | 10.2 | 24.4 | 19.7 | 47.8 | 31.8 |
| | | 99.5 | 64.4 | 67.4 | 75.2 | 75.1 |

Table 1: Probing results comparing Missing- and Complete-State models on GoToLocal. We report accuracy for *Agent-Loc*, and mean Average Precision for *Board-Type/-Color* and *Neighbor-Type/-Color*. For each metric, we show the best performance over all Transformer layers. We observe that compared to Complete-State model which achieves high mAP on reconstructing local state information (i.e. *Neighbor-Type/-Color*), Missing-State model has a better understanding of global state information (i.e. *Agent-Loc* and *Board-Type/-Color*). "Random Init." refers to a randomly initialized sequence model without training.

object types and colors respectively, to account for the class imbalance (e.g., most cells belong to the background). (3) *Neighbor-Type/Color*: We report the same mAP but only consider the $3 \times 3$ neighborhood of the agent. This is to test whether the model can track the change in the neighborhood of agent, which are presumably more important for decision making than other cells on the board.

**Randomly initialized baseline** is a probe trained on a sequence model with random weights. If the trained probes can outperform this baseline, it implies the effectiveness of embodied sequence modeling on the emergence of state representations.

**Initial-state baseline** is to always predict the initial state, without considering the actions taken by the agent. Intuitively, this baseline would perform strongly for *Board-Type* and *Board-Color*, since only the agent and the objects it has interacted with change from the initial states. However, we expect it to perform poorly on inferring agent location and its neighboring objects.

### 5.3 Model Hyperparameters

We instantiate our embodied sequence model with a causal Transformer. It has 6 layers, 8 attention heads, and uses a 768-dimensional hidden size. We train the Transformer model with Adam optimizer. We choose the set of hyperparameters based on validation performances for the corresponding tasks. In the probing experiment, we use a 2-layer MLP probe and optimize it with Adam with a learning rate of 1e-4. All experiments, including Complete- and Missing-State model pretraining and probe training, are performed on a single NVIDIA GeForce RTX 3090. More details of hyperparameters are elaborated in Appendix A.

## 6 Results

We first investigate how different input modalities impact the emergence of abstract state representations. We then evaluate model performance on language-conditioned navigation tasks and demonstrate that Missing-State model performs competitively when compared to Complete-State model which is trained with all available information.

### 6.1 Role of Intermediate State Observations in the Emergence of State Representations

We first perform probing on the internal activations of a pretrained Complete-State model and a pretrained Missing-State model. If a probe can accurately recover certain state properties, we consider the information as being encoded in the internal activations. Note that we expect the internal activations of a pretrained Complete-State model to encode state representations, since it has access to the current states as part of its input.

**Results on GoToLocal.** As shown in Table 1, after pretraining, both Complete-State model and Missing-State model can significantly outperform randomly initialized baselines on probing state properties. For Complete-state model, while the current state is explicitly fed to the model as part of its input, the poor performance of the randomly initialized weights confirms that this information is not trivially preserved in its internal activations – the model needs to be properly pretrained for that to happen. For Missing-State model, even though the current states are not explicitly given, they can be reconstructed reasonably well from the model's internal representations.

In fact, probes of Missing-State model perform better on reconstructing global state information

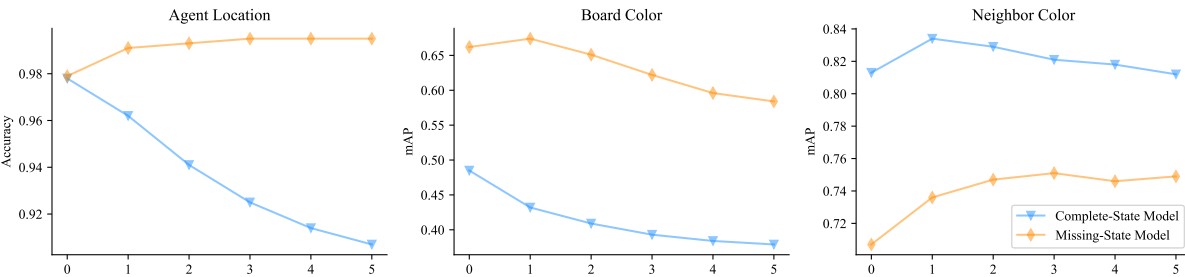

Figure 4: Probing results of the impact of intermediate state observations over Transformer Layers on GoToLocal.

| | Random Init. | Agent-Loc | Board-Type | Board-Color | Neighbor-Type | Neighbor-Color |
|---|---|---|---|---|---|---|
| Initial State | n/a | 19.1 | 78.5 | 85.1 | 34.9 | 24.2 |
| Complete-State Model | ✓ | 19.5 | 26.6 | 32.5 | 49.7 | 35.1 |
| | | 99.7 | 66.9 | 44.4 | 98.5 | 71.9 |
| Missing-State Model | ✓ | 15.8 | 26.3 | 32.5 | 49.9 | 35.2 |
| | | 99.0 | 61.6 | 41.1 | 74.7 | 47.1 |

Table 2: Probing results on MiniBossLevel. We report accuracy for *Agent-Loc*, and mean Average Precision for *Board-Type/-Color* and *Neighbor-Type/-Color*. For each metric, we take the best performance over different Transformer layers. We observe that Missing-State model performs worse than Complete-State model, but much better than the random baseline. We attribute the performance drop from Complete-State to Missing-State to the complexity of MiniBossLevel environment, which makes the emergence of state representations more difficult.

(i.e., *Agent-Loc* and *Board-Type/-Color*) than those of Complete-State model. We speculate one possible reason is that, in Complete-State model, the prediction of next action could "access" the current state which is encoded as part of the input sequence, whereas for Missing-state model, the internal activation of the last action has to encode this information. Perhaps related to this hypothesis, the internal activations (of the last action) in Complete-State model can focus more on the information (e.g., the local neighborhood of the agent) critical for the path planning that complements to the global layout information. This may explain why we observe probes of Complete-State model achieving better performance on local state information reconstruction (i.e. *Neighbor-Type/-Color*).

Figure 4 illustrates probing accuracy over different Transformer layers. For both models, we observe obvious trends on all metrics. In general, from bottom to top Transformer layers, Complete-State model encode less global state information, and Missing-State model encode less local state information but more global state information. One hypothesis for such trends is that the top Transformer layers in Complete-State model mainly work as a policy for decision making, and those layers in Missing-State model mainly works on

inferring the intermediate states.

**Results on MiniBossLevel.** Based on Table 2, we observe that the probes of both pretrained Complete- and Missing-State models can reconstruct the intermediate states reasonably well, in that both outperform the randomly initialized baselines significantly on the probing performance. We find that Complete-State model can capture both local and global state information better than Missing-State model, which we conjecture might be due to the fact that Complete-State model has access to the current state at each timestep, and the environment layout is more complex than GoToLocal. In both GoToLocal and MiniBossLevel, the probes of Missing-State models can effectively recover the intermediate states signicantly better than the random baseline, which demonstrates the emergence of abstract intermediate states in Missing-State models.

## 6.2 Role of Language Instructions in the Emergence of State Representations

To study how language instructions impact the emergence of internal representations of states, we pretrain a Complete-State model and a Missing-State model without passing in language instructions. Since models trained this way cannot perform language-conditioned navigation task, we en-

| | Random Init. | Instruction | *Agent-Loc* | *Board-Type* | *Board-Color* | *Neighbor-Type* | *Neighbor-Color* |
|---|---|---|---|---|---|---|---|
| Initial State | n/a | n/a | 7.4 | 82.2 | 80.1 | 39.8 | 23.2 |
| Complete-State Model | ✓ | ✓ | 10.2 | 27.4 | 21.2 | 47.7 | 31.7 |
| | | | 72.2 | 39.2 | 20.3 | 63.5 | 42.9 |
| | | ✓ | 97.8 | 56.6 | 48.5 | 84.4 | 83.4 |
| Missing-State Model | ✓ | ✓ | 10.2 | 24.4 | 19.7 | 47.8 | 31.8 |
| | | | 88.1 | 44.3 | 21.4 | 62.0 | 40.9 |
| | | ✓ | 99.5 | 64.4 | 67.4 | 75.2 | 75.1 |

Table 3: Probing results showing the impact of language instructions on the emergence of abstract state representations on GoToLocal. We report accuracy for *Agent-Loc*, and mean Average Precision for *Board-Type/-Color* and *Neighbor-Type/-Color*. For each metric, we take the best performance over different Transformer layers. We observe that training without language instructions leads to performance drop on state information recovery. "Instruction" column specifies whether the probed sequence model is trained with language instructions.

sure that both models converge on the training trajectories based on training loss curves. Next, we use the same probing pipeline as Section 6.1 to explore internal representations of states learned by models trained without language instructions.

**Results.** As shown in Table 3, both Complete-State model and Missing-State model trained without language instruction can recover state information significantly better than models without pretraining. This is another piece of evidence that sequence modeling is capable of learning internal representations of states. However, when the language instruction is absent during pretraining, the models struggle to recover state information as accurately as the models trained with language instructions, which reflects that language instructions play an important role in the emergence of internal state representations. We also notice that language instructions are essential for learning object types and colors of the cells. Since this information only occurs in language instructions, the models trained without instructions cannot reconstruct object types or colors of the grid cells as accurately as the models trained with instructions. In all, we observe that language instructions are essential for the emergence of abstract state representations.

### 6.3 Blindfolded Navigation Performance

Experimental results in Sections 6.1 and 6.2 demonstrate that embodied sequence modeling does lead to emergence of internal state representations. We further explore the practical implications of the emergence of internal state representations by evaluating pretrained Complete- and Missing-State models on both GoToLocal and MiniBossLevel.

**Evaluation Benchmark.** We create 108 evaluation tasks for GoToLocal, and 512 evaluation tasks for MiniBossLevel, each task corresponds to a distinct random seed, which defines the environmental layout and the instruction. The number of training trajectories and the number of evaluation tasks are comparable to what were originally used by BabyAI. To ensure that our evaluation tasks test a trained model's generalization behavior, we hold out a subset of objects and a grid sub-region for GoToLocal when creating the training set, and only use them in the evaluation set (see Appendix B.1 for more details). Since MiniBossLevel's task space is much larger, we directly perform random sampling when creating the evaluation set and make sure no duplicate tasks and object layouts appear in the training dataset.

**Evaluation Metrics.** We consider: (1) *Success rate*: the proportion of tasks that the agent is able to accomplish within 64 steps. (2) *Steps-to-Success*: the average number of steps the agent takes when it is able to follow a given instruction successfully.

**Results on GoToLocal.** We train Complete-/Missing- State models with 3 random initializations and report the average performance. We also report the average performance of a random policy over 3 seeds. In Figure 5a, we observe that Missing-State model performs competitively against Complete-State model, even though Missing-State model only has access to the non-initial/last states during its training. This aligns with our expectation that a model which learns internal representations of states can utilize such representations to predict next actions.

**Results on MiniBossLevel.** In Figure 5b, we observe that the success rate of Missing-State

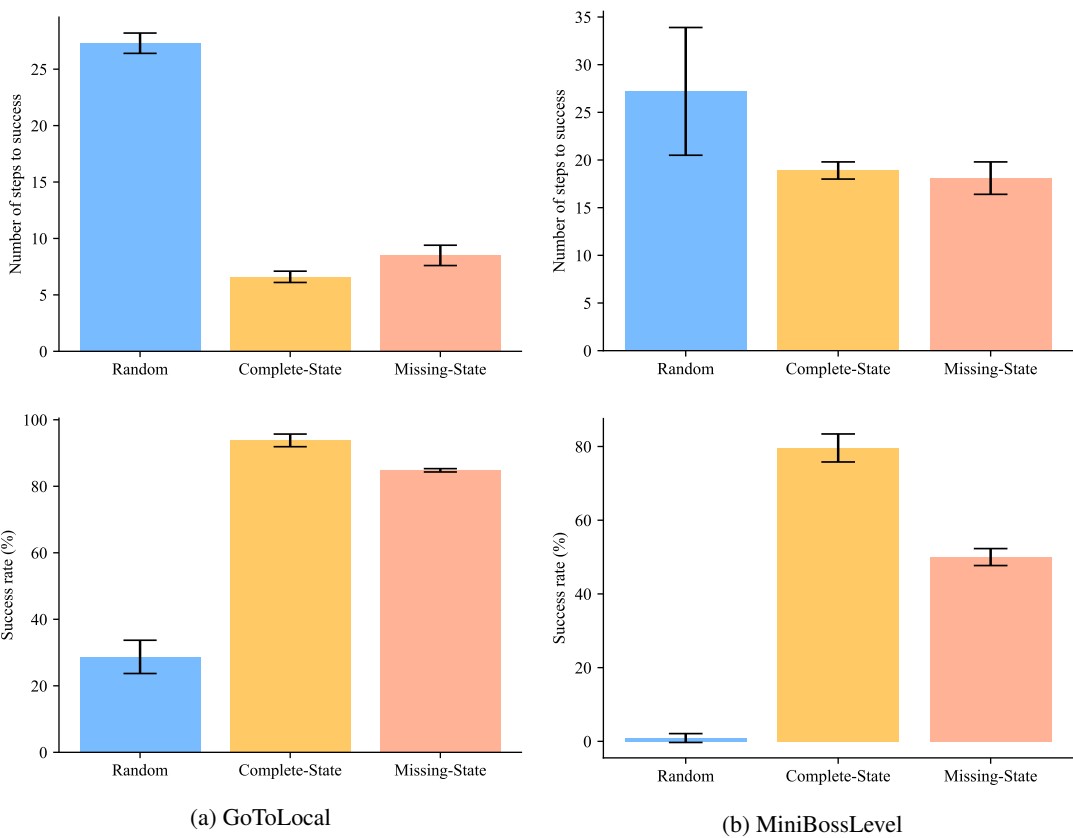

|                | (a) GoToLocal | (b) MiniBossLevel |
| --- | --- | --- |

Figure 5: Evaluation Performance on GoToLocal (a) and MiniBossLevel (b). Higher success rates and fewer steps to success suggest better performance. In both levels, Complete-State and Missing-State models can significantly outperform random policy. We also observe that, without intermediate states, Missing-State models can still achieve competitive performance against Complete-State models on both metrics.

model is lower than that of Complete-State model, while both are significantly higher than the random policy. This is in line with our probing results on GoToLocal where Missing-State model is able to reconstruct intermediate states better than the randomly initialized baseline, but worse than Complete-State model. We also notice that the experimental results reflect MiniBossLevel is a more challenging environment, as the success rate of Complete-State model in MiniBossLevel is significantly lower than that of Complete-State model in GoToLocal. In all, in both GoToLocal and MiniBossLevel, Missing-State models perform competitively against Complete-State models, which serves as further evidence that emergence of internal representations of states are important for next action prediction in emboded sequence modeling.

## 7 Conclusion

In this paper, we provide a series of analyses on whether and how state representations emerge in models trained with the embodied sequence model-ing objective in a language-conditioned navigation task. We find that even when intermediate states (i.e., non-initial/last states) are missing during models' pretraining, the models can still learn to infer the intermediate states, implying that these models automatically learn internal representations of intermediate states. Based on this observation, we further investigate the impact of each input component during pretraining on the emergence of internal state representations for sequence modeling. We observe that language instructions are essential for the models to understand basic affordances of objects and for recovering complete state information. Last, we show the importance of emergent state representations with the finding that the model trained without intermediate states performs competitively against the model trained with complete state information. We hope our work has provided useful insights of the capabilities of sequence modeling for decision making in complex environments and values for future designs of decision making via sequence modeling.

## Limitations

We perform sequence modeling pretraining and probing experiments on GoToLocal and Mini-BossLevel levels. Future work includes evaluating our sequence models in more challenging BabyAI levels or even other environments that simulate real-world settings. Thus, the models need to understand more diverse text-object correlations and infer more involved intermediate states by learning a complex internal world model. While we believe our methodology can be applied to more challenging environments, our architecture assumes language instructions as inputs, which may limit the possible choices of environments.

## Acknowledgements

We would like to thank the anonymous reviewers for their detailed and constructive comments. We are very grateful to Xi Chen, Sebastian Goodman, and Radu Soricut for useful discussions . We appreciate Kenneth Li for helpful feedback on probing experiments and further conversation about potentials of the project. We thank Haotian Fu, Apoorv Khandelwal, Michael Lepori, Calvin Luo, and Jack Merullo for the help on the project. Part of the work was done while Tian Yun was a student researcher at Google Research. This project is in part supported by Samsung Advanced Institute of Technology, and a Richard B. Salomon Faculty Research Award for C.S.

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

# A   Hyperparameters

| Hyperparameters | Values |
| --- | --- |
| # Transformer Layers | 6 |
| Embedding Dim | 768 |
| Symbolic Embedding Dim | 32 |
| # Attention Heads | 8 |
| Nonlinearity | GELU |
| Optimizer | Adam |
| Learning Rate | 3e-5 Complete-State Model
1e-5 Missing-State Model |
| Dropout | 0.1 |
| Epochs | 50 Complete-State Model
100 Missing-State Model |
| Batch Size | 128 |
| Context Size | 64 |
| Max Step Threshold | 64 |
| Max Episode Length | 64 |
| Max Goal Length | 10 |

Table A1: Hyperparameters for pretraining on GoToLocal

| Hyperparameters | Values |
| --- | --- |
| # Transformer Layers | 6 |
| Embedding Dim | 768 |
| Symbolic Embedding Dim | 32 |
| # Attention Heads | 8 |
| Nonlinearity | GELU |
| Optimizer | Adam |
| Learning Rate | 3e-5 Complete-State Model
1e-4 Missing-State Model |
| Dropout | 0.1 |
| Epochs | 50 Complete-State Model
150 Missing-State Model |
| Batch Size | 128 |
| Context Size | 64 |
| Max Step Threshold | 64 |
| Max Episode Length | 64 |
| Max Goal Length | 50 |

Table A2: Hyperparameters for pretraining on Mini-BossLevel

# B   Additional Details and Experiments: *GoToLocal*

## B.1   Evaluation Set Generation

Since GoToLocal environment is relatively simple, 1M training trajectories may cover a significant fraction of different possible configurations, thus leading to leakage from the training set to the test set and hindering the measurement of generalization. Thus, we design a held-out set based on the

| Hyperparameters | Values |
| --- | --- |
| # MLP Layers | 2 |
| Embedding Dim | 768 |
| Learning Rate | 1e-4 |
| Optimizer | Adam |
| Epochs | 200 |
| Batch Size | 128 |

Table A3: Hyperparameters for Probing Experiments

target objects and their locations. Out of 18 color-object combinations, we withhold 3 color-object combinations in training trajectories, but use them in test trajectories. Further, 3 color-object combinations can be placed anywhere in the grid. The remaining 12 color-object combinationswill not be placed in a predefined quadrant in training trajectories, but is allowed to be placed in this quadrant in test trajectories. Based on the held-out rules, we randomly produce a test set of 108 trajectories, where there are 6 trajectories for each of the 18 color-object combinations (i.e., there are 6 colors, and 3 object types for the target object candidates). This set of evaluation trajectories is thus designed to check for compositional generalizability over object types and object colors.

## B.2   Roles of Actions in the Emergence of Abstract State Representations

In Sections 6.1 and 6.2, we have looked into the roles of intermediate states and language instructions in emergent internal representations of states for action sequence modeling. What if we take one more step further by removing everything except actions? Under this scenario, learning to infer intermediate states is quite challenging, since the initial layouts will vary in our language-conditioned navigation task. To deal with this issue, we fix the initial layout and investigate how much state information can be recovered from internal activations of models trained with only actions. We hypothesize language instructions would introduce signals about complete information (i.e., colors and object types) of target objects. To test this, we further explore the scenario when both language instructions and actions are available during the models' pretraining in this fixed layout setting.

**Setup.**   We train two models: *Action-Only model*, which observes only actions, and *Instruction-Action model*, which observes both language in-

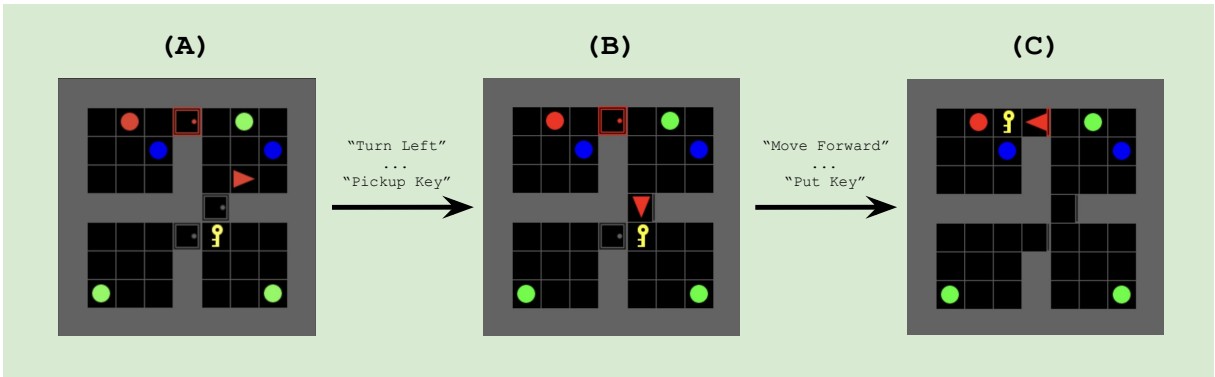

Figure A1: A demonstration of language-conditioned navigation task in BabyAI *MiniBoss* level (Chevalier-Boisvert et al., 2019) where an agent (i.e., red triangle) needs to complete multiple tasks sequentially. This is an example trajectory where the agent achieves the goal of "open the red door behind you and put the key next to the red ball": (A) The board with four rooms is randomly initialized with the agent and several objects. (B) The agent navigates on the board to approach and pick up the key. (C) The agent keeps navigating until it puts the key next to the red ball.

| | Agent-Loc | Neighbor-Occupancy | Neighbor-Object | Neighbor-Color |
|---|---|---|---|---|
| Initial State | 12.1 | 50.0 | 32.8 | 16.7 |
| Action-Only Model | 44.0 | 87.1 | 55.8 | 42.4 |
| Instruction-Action Model | 46.2 | 88.1 | 61.3 | 45.1 |

Table A4: Probing results showing the impact of both actions and language instructions on the emergence of abstract state representations in the fixed layout setting of GoToLocal. We report accuracy for *Agent-Loc*, and mean Average Precision for *Neighbor-Type/-Color/-Occupancy*. For each metric, we take the best performance over different Transformer layers. When no state information is provided, language instructions can still help models learn internal representations of states. Especially, we find the performance gaps are smaller on *Board/Neighbor-Occupancy* and grow larger when the state information to recover is related to object types and colors, which are only included in language instructions.

structions and actions. For model pretraining, we generate 10K trajectories on a fixed initial layout in our language-conditioned navigation task, where each trajectory represents a path from agent's spawn location to a target object. For probing experiments, we generate 1K trajectories as test trajectories. We then randomly sample 10 steps from each training trajectory and 10 steps from each test trajectory as the training and test data [2]. We focus on recovering the cells in agent's 3 × 3 neighborhood (*Neighbor-Type/Color* in Section 5.2) since this is a fixed initial layout setting and recovering the whole grid would be much easier.

**Additional probe metrics.** To understand the role of language, we not only probe for color/object

information for the cells, but also the occupancy of the cells (i.e., whether a cell is occupied by an object, agent, or a wall). We denote this metric as *Neighbor-Occupancy*.

**Results.** Based on the experimental results shown in Table A4, we observe that when state information is completely missing, a model trained with language instructions learns internal representations of states better than a model trained without instructions when it is provided with all types of state information. However, the gaps are small when the state information is related to the basic affordances of objects (i.e., occupancy of cells), but the gaps are significantly large when the state information is related to colors and object types of the cells. This aligns with our expectations, since there is no signal containing the color or object type when the model is trained without language instructions. In all, we find that, in the fixed layout

---

[2]To mitigate the concern of data leakage in training and test sets, we fixed the first 6 actions during the generation of trajectories in test set. With this approach, theoretically, the overlap ratio between training set and test set is $\frac{1}{3^6} \approx 0.14\%$.

| | Agent-Loc | Board-Type | Board-Color | Neighbor-Type | Neighbor-Color |
|---|---|---|---|---|---|
| Complete-State Model - Random Init. | 10.2 | 27.4 | 21.2 | 47.7 | 31.7 |
| Complete-State Model - State Encoder | 100.0 | 73.0 | 78.7 | 85.5 | 84.7 |
| Complete-State Model | 97.8 | 56.6 | 48.5 | 84.4 | 83.4 |
| Missing-State Model - Random Init. | 10.2 | 24.4 | 19.7 | 47.8 | 31.8 |
| Missing-State Model - State Encoder | 26.6 | 41.7 | 46.6 | 46.6 | 33.2 |
| Missing-State Model | 99.5 | 64.4 | 67.4 | 75.2 | 75.1 |

Table A5: Probing results showing the role of Transformer layers in learning abstract state representations on GoToLocal. We report accuracy for *Agent-Loc*, and mean Average Precision for *Board-Type/-Color* and *Neighbor-Type/-Color*. For each metric, we take the best performance over different Transformer layers. We observe that Transformer layers of Complete-State model function as a policy, while those of Missing-State model function to infer intermediate states. "Random Init." stands for randomly initialized baselines. "State Encoder" stands for state encoder baselines.

scenario, language instructions are beneficial for the emergence of understanding basic affordances of objects, and are particularly essential for the emergence of learning grounded information, such as colors and object types.

### B.3 Role of Intermediate State Observations over all Transformer Layers

Figure A3. illustrates the probing accuracy over different transformer layers on all metrics. We observe that probes trained on Missing-State model can achieve higher accuracy on global state recovery (i.e. *Agent-Loc* and *Board-Type/-Color*), while Complete-State model pays more attention to local state information (i.e. *Neighbor-Type/-Color*).

### B.4 Role of the Transformer Block in Embodied Sequence Models

Finally, we validate if the abstract state representations emerge from the Transformer block, or simply from the encoded representation of the observed states. We establish a state encoder baseline, which is a probe trained on the outputs directly obtained from the state encoder (i.e., before passing into Transformer block). Note that the state encoder baseline of Complete-State model takes the corresponding state for each time step as the input; while that of Missing-State model always takes in the initial state as the input for any timesteps.

Table A5 reports results for state encoder baselines on pretrained Complete- and Missing-State models. We start with observations on Complete-State model: compared to the state encoder baseline, the reconstruction accuracy in Transformer block is significantly lower for global board information (Board-Type / -Color), but remains similar for local board information (Neighbor-Type /

-Color). This suggests that the Transformer block retains local information that is critical for predictions, and learns to discard irrelevant information from the encoded states. Note that the state encoder baseline of Complete-State model serves as an "upper bound" since it has access to the per-step state information as its input.

For Missing-State model, the state encoder baseline performs poorly since the state encoder does not have access to the action sequence but only has the initial board information, which is not sufficient to infer the intermediate states. The gains over the state encoder baseline are consistent for global and local states, indicating that (1) the Transformer block encodes intermediate states; (2) simply carrying over the initial state to future steps is not sufficient, even for the global board information.

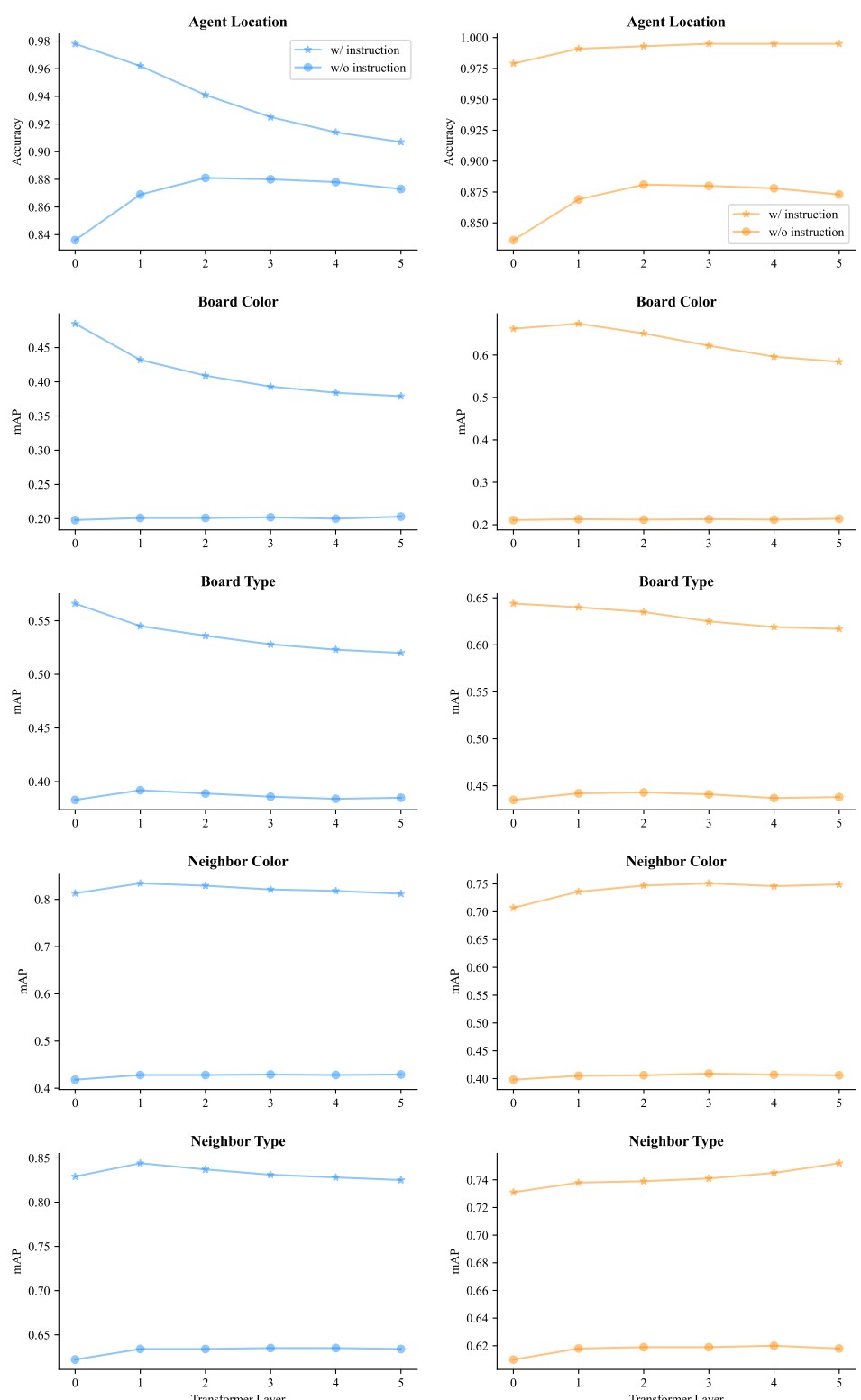

Figure A2: Full Probing Results of the Impact of Instructions over Transformer Layers on GoToLocal. Left: Complete State Model. Right: Missing State Model.

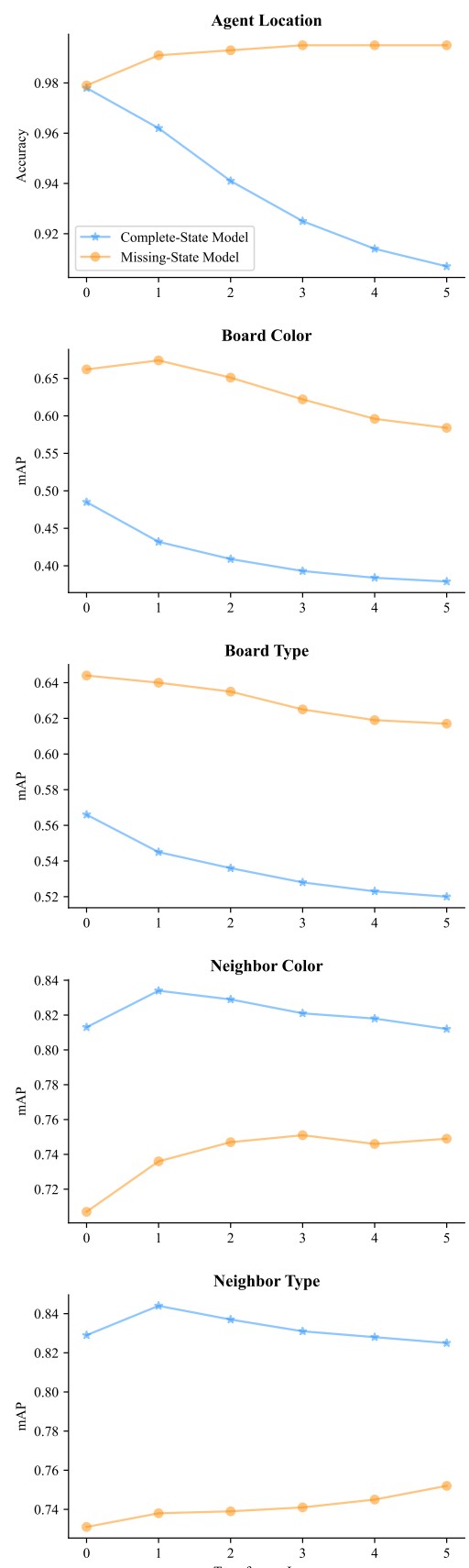

Figure A3: GoToLocal: Probing results of the impact of intermediate states over Transformer Layers on all metrics.