# OpenReview forum: "Emergence of Abstract State Representations in Embodied Sequence Modeling"
_EMNLP/2023/Conference — EMNLP 2023 Main_

### Official Review · Reviewer_GDPh · 2023-07-26

**Soundness:** 3

**Excitement:**

4: Strong: This paper deepens the understanding of some phenomenon or lowers the barriers to an existing research direction.

**Paper Topic And Main Contributions:**

The authors hypothesise that instruction-guided sequence modelling for RL leads to world-grounded internal representations within the agents. To test this they propose a "blindfolded" navigation task in which the agent is given the state of the environment only initially. The authors assume that the model would have to approximate intermediate states -- that would be usually given during the sequence learning itself -- to be successful in this task.

The authors compare Transformer-based sequence learners (i) that have seen intermediate states during training with those (ii) that have not seen intermediate states. They evaluate the agent performances with respect to the ability of probes to reconstruct the environment's layout (colours and objects at each cell). They find that (a) models without the intermediate states succeed about the same amount of times as the others, (b) that giving language instructions is improving the metric (mAP) and (c) that probes on the "intermediate"-conditioned model are better in predicting agent neighbours and probes on the "only initially"-conditioned models are better in predicting global cells.

**Questions For The Authors:**

- Question A: Does the probe prediction included the empty cell? Thus might the probe always predict empty cell (the majority class) and still achieve a high mAP (what would be the value for this behavior)? Having the exact formula used for the mAP would be helpful (or a reference to the utility packaged used).
- Question B: Do you have a single probe for each cell on the grid? (Figure 3 might be misleading, a single probe computing the whole layout?)
- Question C: For the Neighbor prediction, is the agent in the middle of the 3x3 grid?
- Question D: Do you use partial observation or is the agent given the whole grid at each time-step?

**Reasons To Accept:**

- A novel "blindfolded" navigation task that directly aims to test for an agents ability to learn useful intermediate representations of the world (in the minigrid domain).
- A throughout introspection of probing performances in different layers of the Transformer (Figure 4).
- A nice ablation that shows that language instructions impact to probe performances (Table 2).

**Reasons To Reject:**

- The contribution of this work would be stronger when the evaluation had been done for more than a single environment in the minigrid environment. That sequence models are able to learn strong world representations has been already convincingly shown in other work, most famously in the World Models paper of Hu and Schmidhuber (2018) for the Doom and CarRacing environments.
- I find the results (in Table 1) contradict the conclusion that "the models can still learn to infer the intermediate states" (L554). Actually, the results show that the model seems rather sensitive to the state information that is directly passed as an input or not. The Missing-State model achieves higher (non-optimal) scores on "Board" information than on the "Neighbor" information because this information is given initially and carried over to further timesteps. Similarly, the Complete-State model achieves lower scores at global information than on "Neighbor" information because it is highly influenced by the given state at a certain timestep. The more interesting baseline would have been the CNN representations.
- The paper could be more precise in what it actually targets to measure. Of course the model learns "some" useful internal state representation. Are these internal representations the ground truth states? The "internal representation of a state" is only measured via the proxy of probes. Are these internal representations properly grounded? The reported results are not specifically related to the given instruction (Can the agent identify the targeted object?).

**Reproducibility:**

3: Could reproduce the results with some difficulty. The settings of parameters are underspecified or subjectively determined; the training/evaluation data are not widely available.

**Reviewer Confidence:**

3: Pretty sure, but there's a chance I missed something. Although I have a good feel for this area in general, I did not carefully check the paper's details, e.g., the math, experimental design, or novelty.

**Typos Grammar Style And Presentation Improvements:**

- L010: "surface statistics fail to generalize" - I think what is meant is rather the other way around: an agent with a good world model would be able to generalize better. Otherwise I do not fully understand this claim as this is what we usually (successfully) do in deep learning: modelling "surface" statistics (the data). But yes, biases in the data are problematic as these might be exploited by the learning algorithm to achieve a low loss.
- L045: "an embodied" agent -- actually none of the cited or related work is about embodied agents (robots in real or virtual environments; the red triangle is maybe the weakest possible proxy for embodiment) and I would rather connect this work with the term "Offline Reinforcement Learning". But okey, the paper wants to propose this as a new term in L181.
- L072: "pretraining" -- For me it feels that pre-training and training is used not appropriately. I would would rather frame it as "training" the agent and then "evaluate" with probing (instead of "training the probes" L89). Otherwise L177ff "During training" is confusing; same with L277 "trained end-to-end".
- L201: For better understanding, the important term "layout" could be connected with the grid formalism. The term "layout" is actually never properly defined.
- L226: "is computed external" -- is given by the environment
- Figure 2: The display of action space is misleading. Actually there exists only turn left, turn right, move forward in minigrid. But the picture shows "down".
- L264ff: The text could be a bit clearer about "the nature" of the symbols. I guess here it is color and object type. Ok, this is done later in L380.
- L305: "2-layer MLP to reconstruct the current state" -- This could be more elaborate.
- L319ff: "there are 1M trajectories" -- Where are these coming from?
- Chapter 5 reads harder than previous chapters. There are multiple syntax errors, missing "the".
- L410: "use 2 learning rates" -- How's that?
- Table 1+2 could be merged (only having Table 2)

---

> ### Author Rebuttal · Authors · 2023-08-29
>
> ## Response to GDPh
>
> Thank you for your constructive review and also the detailed editorial suggestions. We will incorporate your editorial suggestions in the final version. We respond to your major questions and concerns below.
>
>
> ---
> >**Q1. The contribution of this work would be stronger when the evaluation had been done for more than a single environment in the minigrid environment.**
>
> A1. We agree that evaluating more environments is helpful for supporting our conclusions. In response to your suggestion, we provide additional experiments on a more complex level – MiniBossLevel in BabyAI. MiniBossLevel involves compositional tasks, more language varieties, more objects (e.g, doors), and more actions (e.g., pick up, toggle). We conduct the same experiments (Table 1 and Figure 5) as in our submission and report the results in the tables below.
>
> |                      | Random init. | Agent-Loc | Board-Type | Board-Color | Neighbor-Type | Neighbor-Color |
> |----------------------|:------------:|:---------:|:----------:|:-----------:|:-------------:|:--------------:|
> | Complete-State Model | $\checkmark$ |    19.5   |    26.6    |     32.5    |      49.7     |      35.1      |
> | Complete-State Model |              |    99.7   |    66.9    |     44.4    |      98.5     |      71.9      |
> | Missing-State Model | $\checkmark$ |    15.8   |    26.3    |     32.5    |      49.9     |      35.2      |
> | Missing-State Model |              |    99.0   |    61.6    |     41.1    |      74.7     |      47.1      |
>
>
> We observe that both pretrained Complete- and Missing-State models can reconstruct the intermediate states reasonably well, in that both outperform the _randomly initialized_ baselines significantly on the probing performance. We observe that Complete-State model can capture both local and global state information better than Missing-State model, which we conjecture might be due to that Complete-State model has access to the current state at each time step, and the environment layout is more complex than GoToLocal. Last, we observe that the patterns over Transformer layers also show consistent trends (i.e., increase/decrease over layers) as in GoToLocal (the environment reported in the submission).
>
> We observe that compared to the probing results shown in GoToLocal, the Missing-State model performs worse than the Complete-State model on MiniBossLevel environment on all the probe metrics. One possible cause of this is the much increased complexity of the environment. To further explore the practical implications of the emergence of internal state representations and to better understand the complexity of the environment, we report the agent success rates of the Complete-State and Missing-State models in the table below:
>
> |  | Number of steps to success | Success Rate (%) |
> |----------------------|:--------------:|:------------:|
> | Random policy | 27.2 (± 6.7) | 0.9 (± 1.2) |
> | Complete-State Model | 18.9 (± 0.9) | 79.6 (± 3.8) |
> | Missing-State Model | 18.1 (± 1.7) | 50.0 (± 2.3) |
>
> We observe that the success rate of the Missing-State model is indeed worse than the Complete-State model, while both are significantly higher than the random policy. This is in line with our probing results that the Missing-State model is able to reconstruct intermediate states better than the random baseline, but worse than the Complete-State model. Overall, we observe that the MiniBossLevel is a much more challenging environment, as the success rate of the complete-state model is significantly lower than that of the original environment.
>
> We will update the MiniBossLevel results and visualize the trends over Transformer layers in the appendix of the final version.
>
>
> ---
> >**Q2. That sequence models are able to learn strong world representations has been already convincingly shown in other work, most famously in the World Models paper of Ha and Schmidhuber (2018) for the Doom and CarRacing environments.**
>
>
> A2. The blindfolded setup proposed by our paper studies the **emergence** of world representations via embodied sequence modeling, given only access to the initial state and the action sequences. In contrast, previous work focuses on _explicitly learning_ world representations with a sequence model, assuming full access to the entire states and action sequences of an agent. For example, World Models (Ha and Schmidhuber, 2018) leverages a variational autoencoder to learn a latent state representation via reconstructing each observed state in the training sequences; it then leverages a recurrent neural network to predict the next state representation given the history information.
>
> In short, prior work **explicitly learns** state representations and their dynamics via state reconstruction, whereas we study the **emergence** of state representations via next action prediction (for both regular and blindfolded setups), without using any state reconstruction objectives.
>
>
> ---
> >**Q3. I find the results (in Table 1) contradict the conclusion that "the models can still learn to infer the intermediate states" (L554). Actually, the results show that the model seems rather sensitive to the state information that is directly passed as an input or not. The Missing-State model achieves higher (non-optimal) scores on "Board" information than on the "Neighbor" information because this information is given initially and carried over to further timesteps. Similarly, the Complete-State model achieves lower scores at global information than on "Neighbor" information because it is highly influenced by the given state at a certain timestep. The more interesting baseline would have been the CNN representations.**
>
>
> A3. We would like to clarify that the conclusion that “the models can still learn to infer the intermediate states” (L554) means that Missing-State models can reconstruct the intermediate states significantly better than the randomly initialized baselines. Plus, we treat Complete-State models’ performance as potential upper bounds, since the models can directly access the ground truth intermediate states.
>
> We agree that probing the representations output by the CNN state encoder is another interesting baseline. We established the state encoder baselines for both Missing- and Complete-State models, in which the probes are trained on the outputs from the CNN state encoders (i.e., before passing into Transformer blocks). Note that the state encoder baseline of Complete-State model takes as input the corresponding state for each time step; while that of Missing-State model always takes in the initial state input for any time step.
>
> Below are the tables for state encoder baselines on pretrained Complete- and Missing-State models. Compared with Table 1, for Missing-State model, it performs poorly since its state encoder has no access to the action sequence and solely initial board information is not sufficient to infer the intermediate states. The gains over the state encoder baseline are consistent for global and neighbor states, indicating that simply carrying over the initial state to future steps is not sufficient, even for the global board information.
>
> | | Random Init. | State Encoder Baseline | Agent-Loc | Board-Type | Board-Color | Neighbor-Type | Neighbor-Color |
> |----------------------|:------------:|:----------------------:|:---------:|:----------:|:-----------:|:-------------:|:--------------:|
> | Missing-State Model | $\checkmark$ | | 10.2 | 24.4 | 19.7 | 47.8 | 31.8 |
> | Missing-State Model | | $\checkmark$ | **26.6** | **41.7** | **46.6** | **46.6** | **33.2** |
> | Missing-State Model | | | 99.5 | 64.4 | 67.4 | 75.2 | 75.1 |
>
> For Complete-State model, we observe that the reconstruction accuracy goes down from state encoder baseline to Transformer layers, which reflects that the Transformer layers function as a policy, and they learn to discard irrelevant information from the encoded states. For example, we can see that the global board information (Board-Type / -Color) drops a lot from the state encoder baseline, while the neighbor information remains similar, which implies that Complete-State model learns to localize. Please note that the state encoder baseline serves as an "upper bound" since it has access to the per-step state information as its input. We will include and discuss this set of experimental results in the appendix.
>
> | | Random Init. | State Encoder Baseline | Agent-Loc | Board-Type | Board-Color | Neighbor-Type | Neighbor-Color |
> |----------------------|:------------:|:----------------------:|:---------:|:----------:|:-----------:|:-------------:|:--------------:|
> | Complete-State Model | $\checkmark$ | | 10.2 | 27.4 | 21.2 | 47.7 | 31.7 |
> | Complete-State Model | | $\checkmark$ | **100.0** | **73.0** | **78.7** | **85.5** | **84.7** |
> | Complete-State Model | | | 97.8 | 56.6 | 48.5 | 84.4 | 83.4 |
>
>
> ---
> >**Q4. The paper could be more precise in what it actually targets to measure. Of course the model learns "some" useful internal state representation. Are these internal representations the ground truth states? The "internal representation of a state" is only measured via the proxy of probes. Are these internal representations properly grounded? The reported results are not specifically related to the given instruction (Can the agent identify the targeted object?).**
>
>
> A4. You raise a very interesting point, to which we are sympathetic and which we hope to explore in follow up work. As we understand it, by “properly grounded”, you are asking whether the signal we are detecting via the probes is causally tied to the input and causally implicated in the model’s behavior, as opposed to being “epiphenomenal” so to speak?
>
> Assuming we understand correctly, we agree that probes are a proxy measure. However, the results of probing experiments are still highly informative as 1) they represent a first step which opens the door for future work (i.e., if the probing results are not positive, there is little reason to pursue more advanced methods) and 2) probing classifiers are currently the only standard, widely used method for inspecting internal states. Newer methods are currently being proposed which support causal intervention which we hope to explore/extend in the future. These methods include, for example, INLP [1] and its successors, gradient-based methods [2], and subnetwork/circuit discovery [3,4]. Each of these has potential but is still under active development. Using such methods in our setting warrants a paper in itself, to address all the concerns about these newer methods. Thus, exploring these ideas is done best as a study that follows first showing that the probing experiments yield positive findings (i.e., the current paper).
>
> [1] Ravfogel, Shauli, et al. "Null it out: Guarding protected attributes by iterative nullspace projection." arXiv preprint arXiv:2004.07667 (2020)
>
> [2] Tucker, Mycal, Peng Qian, and Roger Levy. "What if this modified that? syntactic interventions via counterfactual embeddings." arXiv preprint arXiv:2105.14002 (2021).
>
> [3] Conmy, Arthur, et al. "Towards automated circuit discovery for mechanistic interpretability." arXiv preprint arXiv:2304.14997 (2023)
>
> [4] Lepori, Michael A., Thomas Serre, and Ellie Pavlick. "Break it down: evidence for structural compositionality in neural networks." arXiv preprint arXiv:2301.10884 (2023)
>
> ---
> >**Q5. Does the probe prediction include the empty cell? Thus might the probe always predict empty cell (the majority class) and still achieve a high mAP (what would be the value for this behavior)? Having the exact formula used for the mAP would be helpful (or a reference to the utility packaged used).**
>
> A5. Yes, the probe predictions include the empty cells and the empty cells are likely to occupy the majority of the board, which in part makes the output classes imbalanced. Since accuracy would be a misleading metric because of such an imbalance, we use mAP to deal with the skewed distributions of output classes. When the probe always predicts empty cell, it may get high average precision on “Empty Cell” class but much lower scores on the rest of the classes, which eventually leads to a low mAP. Specifically, we use *sklearn.metrics.average_precision_score()* to compute this (i.e., this can be found in the *vlm_rule_learning/model/probe_model.py* of the submitted code as well). We will add this detail to the appendix.
>
>
> ---
> >**Q6. Do you have a single probe for each cell on the grid? (Figure 3 might be misleading, a single probe computing the whole layout?)**
>
> A6. No, we use a single probe (2-layer MLP) for the whole board (or 3x3 neighbor grid). More implementation details can be found in *vlm_rule_learning/model/probe_utils.py* of the submitted code.
>
>
> ---
> >**Q7. For the Neighbor prediction, is the agent in the middle of the 3x3 grid?**
>
> A7. Yes, for the neighbor prediction, the agent is in the middle of the 3x3 grid.
>
> ---
> >**Q8. Do you use partial observation or is the agent given the whole grid at each time-step?**
>
> A8. We use the observation of the whole grid at each time-step.
>
>
> ---
> >**Q9. L410: "use 2 learning rates" -- How's that?**
>
> A9. In the experiments, we observed that the optimal hyperparameters for Complete- and Missing-State models could be different, mainly in terms of learning rate. Thus, we swept different sets of hyperparameters in the pre-training and used the best model checkpoints for probing experiments. We will add the hyperparameters for each model in the appendix and will release the data for pretraining and probing for reproducibility purpose.

---

### Official Review · Reviewer_zk2R · 2023-07-31

**Soundness:** 3

**Excitement:**

4: Strong: This paper deepens the understanding of some phenomenon or lowers the barriers to an existing research direction.

**Paper Topic And Main Contributions:**

This work investigates whether sequence models form representations that are grounded in the environment. Experiments are done on BabyAI environment that combines text instructions and artificial agents that are navigating a gridworld. The authors use a probing technique based on previous work to evaluate the grounding and compare different sequence models (e.g. with and without language).  They found that having instructions was particularly useful for forming grounded representations. Also, they found that the "missing state model" (model that predicts the next actions based on the language instruction, prior actions, and only the initial state) learned representations that were more grounded than representations learned with "complete state model".


**Questions For The Authors:**

What is the dynamics of the emergence of the grounded representation? Do they develop gradually throughout the training? What are the possible benefits of having the grounded representation for this task and could those be quantified (that could be an additional metric that could complement the probing approach that the authors used)? For example, could grounding help with generalization or transfer learning?

**Reasons To Accept:**

The paper is well-written and straightforward. The authors found meaningful differences in terms of grounded representations depending on whether they used instructions or missing/complete state model.

**Reasons To Reject:**

The evaluations are relatively limited and restricted to a single environment. While the authors justified the probing technique used for the evaluation, it would be interesting to have additional ways to explore the grounding (perhaps some evaluation of transfer learning or visualization of the attention layer).

**Reproducibility:**

4: Could mostly reproduce the results, but there may be some variation because of sample variance or minor variations in their interpretation of the protocol or method.

**Reviewer Confidence:**

3: Pretty sure, but there's a chance I missed something. Although I have a good feel for this area in general, I did not carefully check the paper's details, e.g., the math, experimental design, or novelty.

---

> ### Author Rebuttal · Authors · 2023-08-29
>
> ## Response to zk2R
>
> Thank you for your constructive comments. Here, we summarize your major concerns/questions and will incorporate your comments into our final manuscript.
>
>
> ---
> **Q1. The evaluations are relatively limited and restricted to a single environment.**
>
> A1. We agree that evaluating more environments is helpful for supporting our conclusions. In response to your suggestion, we provide additional experiments on a more complex level – MiniBossLevel in BabyAI. MiniBossLevel involves compositional tasks, more language varieties, more objects (e.g, doors), and more actions (e.g., pick up, toggle). We conduct the same experiments (Table 1 and Figure 5) as in our submission and report the results in the tables below.
>
> |                      | Random init. | Agent-Loc | Board-Type | Board-Color | Neighbor-Type | Neighbor-Color |
> |----------------------|:------------:|:---------:|:----------:|:-----------:|:-------------:|:--------------:|
> | Complete-State Model | $\checkmark$ |    19.5   |    26.6    |     32.5    |      49.7     |      35.1      |
> | Complete-State Model |              |    99.7   |    66.9    |     44.4    |      98.5     |      71.9      |
> | Missing-State Model | $\checkmark$ |    15.8   |    26.3    |     32.5    |      49.9     |      35.2      |
> | Missing-State Model |              |    99.0   |    61.6    |     41.1    |      74.7     |      47.1      |
>
>
> We observe that both pretrained Complete- and Missing-State models can reconstruct the intermediate states reasonably well, in that both outperform the _randomly initialized_ baselines significantly on the probing performance. We observe that Complete-State model can capture both local and global state information better than Missing-State model, which we conjecture might be due to that Complete-State model has access to the current state at each time step, and the environment layout is more complex than GoToLocal. Last, we observe that the patterns over Transformer layers also show consistent trends (i.e., increase/decrease over layers) as in GoToLocal (the environment reported in the submission).
>
> We observe that compared to the probing results shown in GoToLocal, the Missing-State model performs worse than the Complete-State model on MiniBossLevel environment on all the probe metrics. One possible cause of this is the much increased complexity of the environment. To further explore the practical implications of the emergence of internal state representations and to better understand the complexity of the environment, we report the agent success rates of the Complete-State and Missing-State models in the table below:
>
> |  | Number of steps to success | Success Rate (%) |
> |----------------------|:--------------:|:------------:|
> | Random policy | 27.2 (± 6.7) | 0.9 (± 1.2) |
> | Complete-State Model | 18.9 (± 0.9) | 79.6 (± 3.8) |
> | Missing-State Model | 18.1 (± 1.7) | 50.0 (± 2.3) |
>
> We observe that the success rate of the Missing-State model is indeed worse than the Complete-State model, while both are significantly higher than the random policy. This is in line with our probing results that the Missing-State model is able to reconstruct intermediate states better than the random baseline, but worse than the Complete-State model. Overall, we observe that the MiniBossLevel is a much more challenging environment, as the success rate of the complete-state model is significantly lower than that of the original environment.
>
> We will update the MiniBossLevel results and visualize the trends over Transformer layers in the appendix of the final version.
>
>
> ---
> >**Q2. While the authors justified the probing technique used for the evaluation, it would be interesting to have additional ways to explore the grounding (perhaps some evaluation of transfer learning or visualization of the attention layer).**
>
> A2. Thank you for suggesting complementary evaluation proxies! We agree that both suggested approaches may be alternative approaches to explore grounding-ness, but both approaches require non-trivial designs. For transfer learning in decision making domain, it is non-trivial to find two tasks which can highlight generalizability of the models. For visualization of the attention weights, previous work (Kobayashi et al. 2020) has shown that the attention weights themselves do not necessarily reflect the importance of tokens.
>
> Compared to the above two proxies, probing is a more direct way to study our research question, as we hope to evaluate whether grounded representations emerged from embodied sequence modeling in a quantitative way. In all, we welcome any of your suggestions on design choices for the evaluation of transfer learning and any further discussions.
>
> [1] Kobayashi et al., 2020. Attention is Not Only a Weight: Analyzing Transformers with Vector Norms. EMNLP 2020.
>
>
> ---
> >**Q3. What is the dynamics of the emergence of the grounded representation? Do they develop gradually throughout the training?**
>
> A3. We provide additional experiments to investigate the dynamics of the emergence of the grounded representation. Below tables show the probing results on intermediate checkpoints throughout the pretraining (i.e., 20% means pretraining with 20% of the total training steps). The bold columns are the best performant checkpoints reported in the submission. We observe that for both models the internal state representations emerge rapidly in the early stage of pre-training, and the probing results saturate as early as 20% of the total steps, subject to some minor fluctuations. Because of the difficulty of “blindfolded” setup, the success rate of Missing-State models converges slower than that of Complete-State models. We also notice the interesting trend that for Missing-State models, the emergence of internal state representations happens at 20%, while the success rate keeps increasing. In general, we observe that the emergence of internal grounded representations is positively correlated with successful task completion.
>
> | Complete-State Model | 0% | 20% | **40%** | 60% | 80% | Final ckpt |
> |----------------------|:----:|:----:|:---------:|:----:|:----:|:----------:|
> | Agent Location | 10.2 | 99.1 | **97.6** | 96.3 | 95.1 | 94.5 |
> | Neighbor Type | 47.7 | 84.0 | **84.4** | 84.6 | 84.4 | 84.3 |
> | Success Rate (valid) | 28.7 | 91.7 | **100.0** | 97.2 | 97.2 | 97.2 |
>
> ---
>
> | Missing-State Model | 0% | 20% | 40% | 60% | 80% | **Final ckpt** |
> |----------------------|:----:|:----:|:----:|:----:|:----:|:--------------:|
> | Agent Location | 10.2 | 99.8 | 99.7 | 99.6 | 99.6 | **99.5** |
> | Neighbor Type | 47.8 | 74.1 | 74.6 | 74.9 | 75.3 | **75.2** |
> | Success Rate (valid) | 28.7 | 72.2 | 72.2 | 86.1 | 83.3 | **91.7** |
> ---
>
>
> >**Q4. What are the possible benefits of having the grounded representation for this task and could those be quantified (that could be an additional metric that could complement the probing approach that the authors used)? For example, could grounding help with generalization or transfer learning?**
>
> A4. Having grounded representations enables the agent to grasp a better understanding of environment layouts (e.g. the obstacles and target objects) given the language instruction, which allows it to plan an optimal path that completes the goals. We agree that “quantifying how much grounding-ness is needed for a task” can help us better understand the importance of grounding. However, to formally study this problem, we would need to define the quantitative metrics to measure the grounding-ness of states and carefully design experiments for evaluations, both of which are non-trivial.
>
> We also believe that the investigation about the relationship between grounding and generalization or transfer learning is a very interesting direction for our future work, which can further demonstrate the significance of grounded representations for generalizability of embodied sequence modeling.
>
> We welcome any suggestions on related works and experimental designs for measuring the grounding-ness needed for a task, or for studying the relationship between grounded representations and generalizability of embodied sequence modeling.

---

### Official Review · Reviewer_89YP · 2023-08-02

**Soundness:** 4

**Excitement:**

4: Strong: This paper deepens the understanding of some phenomenon or lowers the barriers to an existing research direction.

**Missing References:**




**Paper Topic And Main Contributions:**

While sequence modeling for offline RL is highly performant (Janner et al. 2021a; Chen et al., 2021a), it is yet unclear whether deep neural agents trained in this way form grounded representations of their environment, given they are trained to predict "surface statistics" in the sequence modeling objective. The authors investigate whether grounded representations emerge in the sequence modeling task. Here, "emergence of grounded representations" is measured by the ability of an MLP probe to predict the current world state from agents' internal activations. They find that (1) embodied sequence modeling indeed gives rise to grounded representations; (2) inclusion of language instructions in agent input strengthens this emergence; (3) agents given only the initial state and its own action history can represent intermediate world states. That is, sequence modeling objectives indeed give rise to grounded representations.

**Reasons To Accept:**

1. The paper is timely and sheds light on how a relatively new, and potentially widely used, new RL paradigm works.
2. The contribution is easy to understand, the paper very readable, and experiments thorough (e.g., the ablation on language instructions), and the results convincing.

Overall, I find that the contribution of this paper is clear, straightforward, and technically sound.

**Reasons To Reject:**

Overall, I found this to be a strong paper with a straightforward contribution. However, it is missing some references to past work and its underlying motivation can be made more convincing.

1. The underlying motivation can be clarified.
The section "A debate about sequence modeling" can be strengthened. In particular, it's not obvious that the view in Bender and Koller (2020) applies to embodied sequence modeling for decision making. Bender and Koller argue that, in the context of __language modeling__, form prediction without external grounding does not lead to meaning. In the language modeling case, however, there is a crucial separation between what the models are trained on (text corpus; form prediction) and how language is __used__ by humans (in the real world; grounded meaning). In the case of embodied sequence modeling, there is __no clear separation__ between the (pre-)training data and the use; I've understood that the sequence modeling pre-training and actual deployment of the model occur in the same environment. Therefore, I would argue that, intuitively, pre-training RL agents using sequence modeling _on the same environment_ does allow them to capture meaning. In addition, the Bender and Koller (2020) argument has since been refuted; meaning doesn't need grounded referents (Piantadosi & Hill, 2023), and language model representations are able to predict external, visual representations (Ilharco et al., 2021). I think that by saying, e.g., "we take Bender and Koller's view for language/pixel modeling, but test this view empirically for embodied sequence modelling", the motivation can be cleaned up.

2. Missing some key references

- [Systematic Generalization on gSCAN: What is Nearly Solved and What is Next?](https://aclanthology.org/2021.emnlp-main.166) (Qiu et al., EMNLP 2021)

- [Probing Contextual Language Models for Common Ground with Visual Representations](https://aclanthology.org/2021.naacl-main.422) (Ilharco et al., NAACL 2021)

- [A Benchmark for Systematic Generalization in Grounded Language Understanding](https://arxiv.org/abs/2003.05161) (Ruis et al., NeurIPS 2020)

- [Representation Learning for Grounded Spatial Reasoning](https://aclanthology.org/Q18-1004) (Janner et al., TACL 2018)

**Reproducibility:**

4: Could mostly reproduce the results, but there may be some variation because of sample variance or minor variations in their interpretation of the protocol or method.

**Reviewer Confidence:**

3: Pretty sure, but there's a chance I missed something. Although I have a good feel for this area in general, I did not carefully check the paper's details, e.g., the math, experimental design, or novelty.

---

> ### Author Rebuttal · Authors · 2023-08-29
>
> ## Response to 89YP
>
> Thank you for your constructive comments and additional references. Here, we summarize your major concerns/questions and will incorporate your comments into our final manuscript.
>
> ---
> >**Q1. The underlying motivation can be strengthened in the subsection “A debate about sequence modeling” in Section 2. In particular, Bender and Koller (2020) argue in the context of language modeling, which is different from embodied sequence modeling. I would argue that pretraining RL agents using sequence modeling on the same environment does allow them to capture meaning. I think that by saying, e.g., “we take Bender and Koller’s view for language/pixel modeling, but test this view empirically for embodied sequence modeling”, the motivation can be cleaned up.**
>
> A1. Thank you! We appreciate you pointing this out, and agree that the current wording is not a fair characterization of that debate. We will reword the section titled “a debate about sequence modeling” to read as follows:
>
> *Though sequence modeling has been shown to be successful in language modeling [1, 2], image modeling [3], and decision making [4, 5, 6], it remains unknown whether such training results in models that encode rich abstractions or rather simply model surface statistics of their domains. Recent work has suggested that sequence modeling of language tokens leads to representations of abstract concepts such as syntax [7, 8] and even non-linguistic notions such as of colors and directions [9]. Such findings suggest that sequence modeling over embodied domains (like we study here) could similarly result in abstractions over states and actions.*
>
> *A related debate concerns whether sequence modeling over text is sufficient to encode meaning [10]; however that work is concerned with the dependence on text, not with sequence modeling itself. Thus, our work is related but makes a complementary point, namely, that sequence modeling over non-text data can lead to good abstractions over the non-linguistic world.*
>
> [1] Raffel, Colin, et al. "Exploring the limits of transfer learning with a unified text-to-text transformer." JMLR 2020.
>
> [2] Sanh, Victor, et al. "Multitask prompted training enables zero-shot task generalization." arXiv preprint arXiv:2110.08207
>
> [3] Dosovitskiy, Alexey, et al. "An image is worth 16x16 words: Transformers for image recognition at scale." arXiv preprint arXiv:2010.11929
>
> [4] Janner, Michael, Qiyang Li, and Sergey Levine. "Offline reinforcement learning as one big sequence modeling problem." NeurIPS 2021.
>
> [5] Chen, Lili, et al. "Decision transformer: Reinforcement learning via sequence modeling." NeurIPS 2021.
>
> [6] Li, Shuang, et al. "Pre-trained language models for interactive decision-making." NeurIPS 2022.
>
> [7] Hewitt, John, and Percy Liang. "Designing and interpreting probes with control tasks." arXiv preprint arXiv:1909.03368
>
> [8] Tenney, Ian, et al. "What do you learn from context? probing for sentence structure in contextualized word representations." arXiv preprint arXiv:1905.06316
>
> [9] Patel, Roma, and Ellie Pavlick. "Mapping language models to grounded conceptual spaces." ICLR 2021.
>
> [10] Bender, Emily M., and Alexander Koller. "Climbing towards NLU: On meaning, form, and understanding in the age of data." ACL 2020.
>
>
> ---
> >**Q2. Missing some key references.**
>
> A2. We appreciate these additional references and will incorporate them into our camera-ready.

---

### Meta-Review · Area_Chair_vtvH · 2023-09-19

**Recommendation:** 4

**Metareview:**

The paper is written well and clear to read. The introduced problem and method are novel, and the experiments are thorough. The related work could be updated with some references suggested by the reviewers. The paper would also be stronger if it considered more than one environment for evaluation.

---

### Decision · Program_Chairs · 2023-10-07

**Decision:**

Accept-Main

**Comment:**

The paper is written well and clear to read. The introduced problem and method are novel, and the experiments are thorough. The related work could be updated with some references suggested by the reviewers. The paper would also be stronger if it considered more than one environment for evaluation.